# OpenReview forum: "Game-Theoretic Multi-Agent Collaboration for AI-Driven Scientific Discovery"
_ICLR.cc/2025/Workshop/AgenticAI — ICLR 2025 Workshop AgenticAI Reject_

### Official Review · Reviewer_VZ9F · 2025-03-02
**Lack of Intellectual Engagement and Scientific Rigor**

**Rating:** 2
**Confidence:** 5

**Review:**

Summary:

This paper proposes a game-theoretic multi-agent AI framework for scientific discovery, where AI agents negotiate hypotheses and manage computational resources using Nash equilibrium and cooperative game theory. The authors claim that their approach has been tested in climate modeling, astrophysics, and biomedical research, demonstrating improvements in hypothesis validation and resource efficiency. However, the manuscript primarily consists of bullet points rather than fully developed text, raising serious concerns regarding its originality, coherence, and scientific rigor.

Weaknesses:

1. The manuscript is largely composed of bullet points, lacking detailed explanations and complete sentences. This severely impacts readability and prevents a thorough assessment of its depth, clarity, and scientific contribution.

2. Several critical components typically expected in a scientific paper are either missing or ineffective, including a well-structured Introduction, a clear Problem Statement, a thorough Related Work review, detailed Implementation and Experimental Methods, and a meaningful Result Analysis. The absence of these sections gives the impression of minimal human effort and intellectual engagement.

3. A manual verification of citations revealed significant discrepancies. For instance, the first reference in the bibliography— Zhu, Y., Morales, S., & Kim, M. (2021). Single-Agent vs. Multi-Agent Paradigms in Scientific AI. ACM Surveys on AI, 4(2), 101–120.—could not be found despite extensive searching. Furthermore, the cited venue, ACM Surveys on AI, appears to be non-existent, raising serious concerns about the validity and authenticity of the references.

Overall, this paper raises significant concerns regarding potential AI-generated content. Even if human-authored, it suffers from fundamental flaws in theoretical grounding, empirical validation, and scientific contribution, making it unsuitable for publication.

---

### Official Review · Reviewer_NATZ · 2025-03-04
**Game-Theoretic Multi-Agent Collaboration for AI-Driven Scientific Discovery**

**Rating:** 3
**Confidence:** 3

**Review:**

This paper presents a game-theoretic multi-agent AI framework for scientific discovery, where autonomous agents negotiate and refine hypotheses in competitive and cooperative settings. By incorporating Nash equilibrium and cooperative bargaining, the system effectively optimizes resource allocation and conflict resolution across diverse domains such as climate modeling, astrophysics, and biomedical research. However, the paper lacks clarity and depth, requiring more comprehensive details in each section. A more detailed definition of the Comparison Metrics will be helpful to better understand it. The Performance Analysis section needs a more thorough discussion to strengthen the evaluation. Additionally, while the framework enhances scientific collaboration, it faces challenges such as high computational complexity, uncertain synergy estimation, and scalability limitations beyond 50 agents.

---

### Official Review · Reviewer_7U8n · 2025-03-05

**Rating:** 4
**Confidence:** 4

**Review:**

This paper presents a game-theoretic multi-agent AI framework for scientific discovery, where autonomous AI agents collaborate or compete to generate and refine hypotheses. The system models resource allocation and hypothesis validation as strategic games, using Nash equilibrium for competitive scenarios and cooperative bargaining for collaborative ones. The approach is applied to climate modeling, astrophysics, and biomedical research, showing that it improves resource efficiency, scientific output, and negotiation stability. Experimental results demonstrate that this framework enhances computational resource sharing, accelerates hypothesis validation, and reduces conflicts in decentralized scientific collaborations.

Strengths of the Paper:
1. Provides a structured approach to resolving conflicts in scientific collaborations and uses both competitive and cooperative strategies, adapting to different research scenarios.

Weaknesses of the Paper:
1. The paper is not written properly. The discussion is very brief.
2. Computational complexity increases with more agents, making large-scale applications difficult.

---

### Decision · Program_Chairs · 2025-03-05

Reject